# The Empathetic Involvement of Nurses in the Context of Neuroscience: A Mixed-Methods Study

**DOI:** 10.3390/healthcare12202081

**Published:** 2024-10-18

**Authors:** Antonio Bonacaro, Federico Cortese, Chiara Taffurelli, Alfonso Sollami, Cinzia Merlini, Giovanna Artioli

**Affiliations:** 1Department of Medicine and Surgery, University of Parma, 43126 Parma, Italy; antonio.bonacaro@unipr.it (A.B.); federico.cortese@unipr.it (F.C.); chiara.taffurelli@unipr.it (C.T.); cinzia.merlini@unipr.it (C.M.); 2University Hospital of Parma, 43126 Parma, Italy; alfonso.sollami@unipr.it

**Keywords:** empathy, emotion regulation, nurses, patient outcomes, neuroscience nursing, mixed methods

## Abstract

**Background/Objectives:** Empathy and emotional regulation (susceptibility and resistance) play an important role in a nurse’s well-being and the provision of high-quality care. This phenomenon has not yet been studied in the context of nurses working in neuroscience. This study aimed to explore the perceptions related to empathy among nurses working in neuroscience contexts. **Methods:** Employing a mixed-methods approach, we conducted an online quantitative survey with 211 nurses working in various neuroscience settings using the Balanced Emotional Empathy Scale (BEES) and 15 online semistructured qualitative interviews to delve deeper into empathetic experiences. The mean and measures of dispersion, such as standard deviation, were used to analyze the quantitative data. Thematic analysis investigated qualitative data, and data triangulation was performed. **Results:** The quantitative findings revealed no significant differences in empathy or emotional regulation across the different neuroscience settings but highlighted an increase in susceptibility related to young age (under 29) and years of service (first 5 years). The interviews brought to light the challenges nurses face in highly emotional situations and the strategies they employ to manage empathy and maintain professional detachment, such as self-care strategies, awareness development, and team support. One hindering factor is managers. **Conclusions:** The findings of this study underscore the essential role of empathetic capability in nursing care in neuroscience. The experience of younger nurses and the first 5 years of employment are elements to be considered by managers for burnout risk. Nurses demonstrate susceptibility and resistance and maintain a balance in dealing with high-emotional-stress situations. The implications of these findings are significant and should guide future research and practice in the field of neuroscience nursing.

## 1. Introduction

Empathy is a fundamental aspect of healthcare, particularly in nursing care, where the ability to understand and share patients’ emotions is crucial for protecting themselves as health professionals and providing high-quality care. Empathy has been extensively studied in various fields [1,2] and health professions, especially for those in direct contact with patients and families suffering.

In this study, we intend to measure the empathy of nurses working in facilities that house patients with neurological pathologies in Italy. Since empathy is a complex phenomenon, we intend to evaluate it in extension and depth. Based on this rationale, we have chosen a mixed-methods methodology that integrates quantitative and qualitative methodologies [3,4].

This study is driven by a pivotal research question: What are the perceptions of empathy among nurses working in neuroscience contexts? Specifically, (i) what is the relationship between resistance and susceptibility in neuroscience nurses? Furthermore, (ii) what do Italian nurses in neuroscience have to say about their empathetic experiences with their patients?

To answer the first subquestion, we used the Balanced Emotional Empathy Scale (BEES) [5]. The scale measures two variables: susceptibility and resistance, which are considered fundamental variables of the emotional regulation activated when health professionals assist patients and families with high levels of suffering. The choice of the BEES is related to the fact that, although it is a fairly dated scale, it has been validated, translated, and adapted to Italian culture and is already validated in this context [5].

The scale was emailed to members of the Italian Association of Nurses in Neuroscience (ANIN) because we are interested in measuring the phenomenon extensively, reaching as many nurses as possible to apply statistical analyses that can indicate significance.

To answer the second subquestion, the research team was interested in exploring the concept of empathy in-depth in the context of Italian neuroscience nurses by listening to the voices of those concerned. It was decided to integrate a qualitative part into this study through semistructured interviews with several professionals related to the association’s members until data saturation. The interview track was developed from the questionnaire, with questions, for example, related to responsiveness to emotional stimuli. In the interview, questions about professionals’ real experience of empathy were integrated [4].

The data from the questionnaires were analyzed by statistical analyses.

Interview data were analyzed by thematic analyses to highlight the main issues [6].

A mixed-methods study was developed, with triangulation of the results: quantitative and qualitative data were compared to see if there were convergences or divergences between them [3].

### 1.1. Background

#### 1.1.1. Empathy in Healthcare

Neuroscience studies have identified specific brain regions, such as the anterior insular cortex and the cingulate cortex, that are linked to empathy, particularly in the perception of pain [7,8]. This connection between the neural mechanisms of empathy and the experience of pain underscores the intricate relationship between emotions and physiological responses [9]. Several studies have explored the different neuroscientific mechanisms of empathy, its development, and its usefulness in healthcare services [10].

Empathy, understood as the ability to understand and share the emotions of others, has been extensively studied for its impact on therapeutic relationships and care outcomes [11,12,13,14].

Maintaining high levels of empathy in clinical practice is essential for healthcare professionals, as it can influence patient outcomes and satisfaction [9]. A recent study in China involving 100 nurses recruited from both hospital and community settings demonstrated that there is variation in the perception of empathy related to patient pain between the hospital and home environments and regarding the length of service, activating neurocognitive mechanisms differently. Nurses who have worked longer have fewer negative perceptions of pain in hospitals, while perceiving more pain in patients at home. The modulation of empathy represents an adaptive advantage for the professional in different work contexts [9].

Research has shown that empathy has a significant impact on the nurse–patient relationship, with empathic interactions leading to positive therapeutic outcomes and greater ethical awareness among nurses [15]. This was particularly evident during the COVID-19 period, where the psychological impact and empathy conveyed to patients were especially important [16,17].

However, attention is also shifting to the challenges nurses face in exercising empathy, particularly the risk of burnout and managing their own emotions when faced with patients with critical health conditions [18,19].

Studies like that of Lakemann et al. [20] suggest that empathy is not static but can be developed and modulated through education and reflective practice. Moreover, the work of Awe et al. [21] underscores the importance of the work environment and organizational support in facilitating or hindering the expression of empathy, indicating that management practices can have a significant impact on nurses’ emotional well-being. Mindfulness and awareness have emerged as promising strategies to help nurses manage the emotional demands of their work, promoting greater emotional stability and an increased capacity to maintain sustainable empathic engagement [18,22].

#### 1.1.2. Empathy in Neuroscience Nursing Setting

Nursing practice in neuroscience settings is at the intersection of advanced clinical demands and unique emotional challenges, where empathy plays a crucial role in defining the quality of care and the well-being of healthcare staff.

In the area of neuroscience, in brain tumors, patients express a need for more information from nurses to deal with the time of diagnosis, and there are still many unmet needs, such as communication and support needs, that are not adequately taken into account by health professionals [23,24]. Nurses who address these needs using therapeutic communication contribute to improving the quality of life for patients and families [25].

Neuro-oncology patients highly appreciate the nurse’s role, which positively influences the quality of life of patients and family members [26].

Families and patients with motor neuron disease, multiple sclerosis, or Parkinson’s disease, as well as people with dementia, express the need for support in different stages of their disease, especially at the end of life when the need to integrate palliative care into neurological care becomes urgent [27]. For end-of-life needs, communication is an essential element. However, even here, there are few studies that address the professional patient and family relationship and provide support to families of patients with ALS [28,29].

Recent studies explore the quality of life not only of patients and family members but also of professionals, especially doctors and nurses, who work constantly in contact with patients with multiple sclerosis or ALS. Nurses express a high risk of compassion fatigue, especially in the first years of work [30]. Although the attention of professionals is increasing, studies have yet to investigate this phenomenon [31].

Empathy and compassionate care are recognized as essential competencies of professionals to provide high-quality care [32,33].

Empathy is also recognized as a “double-sided mirror”, since it can promote a better quality of life for patients and families and better well-being for professionals who encounter strong emotions [34,35].

Empathy is therefore studied in individual neurological pathologies but not across the neuroscientist’s nursing.

#### 1.1.3. Measuring Emotion Regulation in Healthcare

The dimension of emotion regulation is a subject of recent studies in the health professions, especially when they continuously encounter highly stressful situations. The literature suggests activating strategies that promote emotional regulation, thus developing greater resilience in facing adverse events [36,37,38].

In the field of emotional regulation, two fundamental domains intervene: susceptibility and resistance. Sensitivity or susceptibility refers to the ability or tendency to be affected by emotional feelings. Emotional resistance implies the avoidance of feelings and unconscious emotions hidden under the surface. Measuring resistance and susceptibility in healthcare professionals is useful to better understand the phenomenon of emotional regulation and the ability of workers to prevent distress and enhance personal and professional well-being, as well as to feel empathy or detachment from patients and their families. To evaluate these dimensions, BEES represents a validated and effective scale. Its validation in Italian has shown that its factorial structure is at least partly superimposable with the Emotional Empathic Tendency Scale. A translation was carried out that also considers the Italian culture and way of perceiving emotions. The 30 items of the scale are grouped into two domains, resistance and susceptibility [5]. Many studies have investigated empathy in healthcare professionals [13,39].

Various studies have investigated the domains of resistance and susceptibility in emotion regulation. Of these, few are aimed at nurses, and there are no studies aimed at neuroscience nurses [40,41].

#### 1.1.4. Mixed-Methods Research

Mixed-methods research intentionally combines perspectives, approaches, forms of data, and analyses associated with quantitative and qualitative research to develop nuanced and comprehensive understandings [3].

The advantages of a mixed-methods approach are (i) that it involves a complex and sophisticated methodology using both quantitative and qualitative data and (ii) that it allows information to be drawn from beyond the quantitative and qualitative analysis of the data. The main disadvantage is that it requires researchers to have quantitative and qualitative research skills [4].

Data integration, the cornerstone of mixed methods, is the explicit conversation between the quantitative and qualitative components of a study [42]. The unique challenge of mixed methods lies in planning and implementing this meaningful integration, which will engage and intrigue researchers, academics, and students interested in research methodologies [43].

Integration serves to provide new information about the subject of study and can be performed with diagrams that visually show the points of integration and with practical tables or graphs to help achieve a meaningful integration in mixed-methods studies [44,45].

Three methods of integrating data are indicated in the literature: (i) the project of convergent mixed methods, (ii) the project of explanatory sequential mixed methods (first quantitative and then qualitative), and (iii) the project of explanatory sequential mixed methods (first qualitative and then quantitative) [3].

In this study, we use a mixed-methods convergent design involving the implementation of independent, quantitative, and qualitative study components. Results are first presented separately and then integrated into the data triangulation by merging analysis and interpretation [3].

Hirose, M., and Creswell, J. W (2023) identify the following six quality criteria for mixed methods: (i) advance a rationale for using a mixed-methods methodology [4]; (ii) write quantitative, qualitative, and mixed questions or objectives [46]; (iii) report quantitative and qualitative data separately [47]; (iv) name and identify the design type of the mixed methods approach and present a diagram of it [45]; (v) declare the use of integration in a joint view; and (vi) discuss the meta-inferences and value resulting from the integration analysis.

We tried to meet the quality criteria for mixed methods in our article.

This study aimed to evaluate empathy among nurses working in neuroscience contexts. Specifically, we measured the domains of susceptibility and resistance in nursing neuroscience contexts (quantitative study). Furthermore, we explored, in-depth, the empathetic experiences of nurses in neuroscience with their patients in clinical practice (qualitative analysis).

## 2. Materials and Methods

### 2.1. Study Design

Adopting a mixed-methods approach [48], this study combined qualitative and quantitative methods to gain a comprehensive understanding of nurses’ perceptions of empathy. The research was conducted in February 2022, over a period of twenty days for the questionnaire and a period of ten days for the interviews, aiming to capture both the measurable and experiential dimensions of empathy within the nursing context.

We used the online questionnaire in the first instance and then the interviews.

### 2.2. Quantitative Data Instruments

The quantitative component utilized the Balanced Emotional Empathy Scale (BEES), validated in Italian [5], to assess the propensity for empathy. The questionnaire, consisting of 29 items in its Italian version, was administered through a self-completed and anonymous format, using a 7-point Likert scale. This psychometric scale consists of 30 items (in the original version) and is employed to assess empathic capacity, specifically how intensely an individual emotionally responds to the emotions of others. The validation study of the BEES scale [5] revealed the presence of 2 macroclusters (item groupings), consisting of the union of positively worded items on one side and the union of negatively worded items on the other. This prompted the researchers to begin the validation study by starting with a confirmatory principal component analysis that extracted only two dimensions [5]. The two factors investigated are “resistance” and “susceptibility” to emotional stimuli. “Resistance” refers to imperviousness to contagion from internal emotional states. “Susceptibility” means susceptibility to contagion from emotional states [5].

Each item on the scale presents a potentially emotional situation, asking the respondent to indicate their level of emotional empathy through a 7-point Likert scale, ranging from “strongly disagree” to “strongly agree”. This response method allows for a detailed and nuanced exploration of empathic reactions, reflecting individual variability in emotional responses [5].

The choice of this instrument is based on its reliability and validity, proven by numerous studies that attest to its effectiveness in various contexts, making it an ideal tool for measuring empathy in a clinical setting. By using the BEES, this research was able to provide an objective measure of empathy, essential for better understanding the empathic behavior of nurses and its implications in patient care.

Quantitative data were collected using an online survey through the network of the National Association of Neuroscience Nurses, following a request for consent included in the questionnaire.

The questionnaire took about 15 min to complete and was administered before the interviews.

### 2.3. Qualitative Data Instrument

After the online questionnaire was administered, we sent an e-mail to the ANIN membership network to ask for their cooperation in an online interview as well. The participants responded spontaneously. The interviews were conducted individually, and our interview length assumption was 20–30 min.

The interview questions were formulated using the BEES items through a discussion between the researchers.

The qualitative aspect was explored through semistructured interviews using the ZOOM platform to facilitate data collection in compliance with privacy regulations and informed consent. Researchers formulated questions about the following areas of interest:Empathic experiences in the workplace.Responsiveness to emotional stimuli.The evolution of one’s empathic skills.Sociodemographic questions (nationality, age of the nurse, gender, education, how long they have been in the profession, professional title, post-basic training, years of service in the current unit, shift work).

The interview concluded with final questions, such as “Do you have anything else to add, or any questions before we conclude the interview?”. These questions allowed for the introduction of any concerns or negative aspects the participant might have thought of or that might worry them.

The convergent triangulation design allowed for the integration and discussion of results obtained from the two research methods [49], providing a comprehensive and articulated view of nursing empathy in neuroscience.

### 2.4. Setting and Sampling

The setting included neurosciences departments, such as the neurointensive care unit (ICU), neurosurgery, mental health, neurology, neurorehabilitation, neuroradiology, and their pediatric counterparts.

The sample, selected through a convenience approach, included nurses active in neuroscience departments or registered with Associazione Nazionale Infermieri Neuroscienze (ANIN) in the years 2020–2022 and providing consent to the online questionnaire or to the interview.

For quantitative data, sample size was calculated using a 95% confidence interval and a standard error of 5%, out of a total population of 350 nurses; the result is 183 respondents.

For qualitative data, we used the saturation criterion.

The inclusion criteria were, for both the questionnaire and interview, only neuroscience nurses registered with the Italian association and providing consent to complete the online questionnaire. The exclusion criteria were not operating in neuroscience settings and not providing consent for this study.

Participants were chosen based on their availability and accessibility, ensuring the representativeness of the different neuroscience areas.

The research team used the association’s mailing list (ANIN) to send the questionnaire online to all registered nurses (N°350).

The socioanagraphical data we wanted to collect were age, level of education, neuroscience area, total years of work, and years in the neuroscience field.

Through the network of the national association, nurses came from different institutes and regions.

### 2.5. Data Analysis

#### 2.5.1. Quantitative Data

The quantitative collected data were statistically analyzed using SPSS^®^ 28.0 software. Measures of central tendency, particularly the mean, and measures of dispersion, such as standard deviation, were employed to describe the collected data. The analysis focused on the following aspects:BEES scale factors.Individual scale items.Participants’ age and years of service.

This quantitative approach [6,50] allowed for the measurement of nurses’ empathy tendencies, providing a solid foundation for further qualitative investigations.

#### 2.5.2. Qualitative Data

Thematic analysis was the chosen methodology for processing the qualitative data derived from the semistructured interviews. This process involved deconstructing and subsequently reconstructing the data to identify recurring themes and significant patterns. The identified content categories were divided into thematic areas or codes, which guided data interpretation [6,50].

The researchers used the criterion of data saturation to define the number of interviews to be carried out, in accordance with the definition reported by [51], in which saturation “relates to the degree to which new data repeat what was expressed in previous data”.

The interviews were analyzed as they were conducted. At the twelfth interview, the data were saturated. Three more interviews were conducted to confirm saturation.

We inserted data that were coded manually by two researchers independently. The results were compared to find agreement. In case of disagreement, a third researcher intervened.

Approximately 300 initial codes were identified, which were then processed according to the methodology of the Braun and Clark thematic analysis [6,50].

Thematic analysis is a method for identifying, analyzing, and reporting patterns (themes) within data. This method involved six steps: Phase 1: familiarizing oneself with one’s data; Phase 2: generating initial codes; Phase 3: searching for themes; Phase 4: reviewing themes; Phase 5: defining and naming themes; and Phase 6: producing the report.

The research team tried to adhere to the following criteria in order to provide rigor to our work: reliability, validity, and generalizability [52].

The combination of quantitative and qualitative analysis, through the design of convergent triangulation [49], allowed for a holistic interpretation of the data, enriching the understanding of the empathic phenomenon among nurses in neuroscience contexts.

The data mentioned in the following text corresponded to the collection of quantitative and qualitative data.

### 2.6. Ethical Statement

This study was developed according to the Helsinki guidelines, with informed consent requested from the participants and data handled in an aggregated and anonymous manner. The Executive Board of the National Association of Neuroscience Nurses (NANA), Italy, approved this study with protocol number 2022_000003, year 2022.

## 3. Results

### 3.1. Quantitative Findings

The survey involved a sample of 211 nurses working across various specializations in the field of neuroscience, registered with the ANIN; average age: 41.1, SD 12.5; average seniority age: 17.02, SD 12.6; average seniority age in neuroscience context: 7.44, SD 8.76.

The sociodemographic data collected showed a diverse distribution of nurses across different operational units, reflecting the complexity and heterogeneity of the field of neuroscience nursing. Table 1 lists the operational units of the participants and the distribution in the total sample.

The use of the Balanced Emotional Empathy Scale (BEES) allowed for the assessment of the participants’ empathic tendencies, dividing the results into two main factors: “Resistance” and “Susceptibility”. The Cronbach Alpha values obtained, 0.779 for Resistance and 0.876 for Susceptibility, confirm the reliability of the scale in our study population. An ANOVA analysis explored the differences between the operational units of the participants, revealing no statistically significant differences, suggesting that nurses’ empathic tendencies remain substantially consistent across different specializations. Table 2 lists the averages of the two factors analyzed (resistance and susceptibility).

However, interesting variations were observed when data were analyzed in relation to total years of service. Specifically, the “Empathy” factor showed significant differences, indicating how accumulated experience might influence nurses’ ability to emotionally connect with patients.

Significance also emerged between some sociodemographic variables and the BEES scale. A high level of susceptibility was detected in nurses under 29 years of age (M = 5.79), which lowers around the age range of 31 to 40 years (M = 5.14) and then slightly rises towards 42 years of age (M = 5.31). Figure 1 shows the trend of susceptibility and resistance based on registry age.

These data conform with the data that emerged from years of service; in fact, nurses at the beginning of their career have a higher degree of susceptibility (M = 5.84), which drops drastically around six years of service (M = 5.26). Figure 2 shows the trend in susceptibility and resistance based on years of service.

Confirmatory factor analysis (CFA) revealed the following: chi-square test: X^2^(349) = 707; *p*-value: *p* < 0.001; Tucker and Lewis Index: TLI = 0.751; Root Mean Squared Error of Approximation: RMSEA = 0.0697 (IC95%: 0.0623–0.0771). Bayesian Information Criterion: BIC = 22314; and Akaike Information Criterion: AIC = 22029

The values showed a fairly good fit of the model with the two factors.

### 3.2. Qualitative Findings

Fifteen nurses were interviewed. Semistructured interviews provided valuable insights into the empathic experiences of nurses in the neuroscience context. The nationality was Italian. The average age of the participants was 38.6. Total seniority recorded had an average of 17.9 years, while seniority in the neuroscience area had an average of 10.8 years. The average interview duration was 33.5 min. Table 3 shows the details of the individual interviews, which vary widely from a minimum of 20 min to a maximum of 55 min. Table 3 lists the sociodemographic characteristics of the participants in the qualitative research.

Participants shared moments of deep emotional involvement, often related to situations of significant patient vulnerability, such as accompanying patients in the terminal phase of life, assisting young patients with adverse prognoses, and managing particularly difficult news.

From the narratives, the importance of empathy as a care tool emerged, allowing nurses to better understand the needs and emotions of patients and their families. However, this emotional involvement is not without challenges: many nurses discussed the difficulty of maintaining a balance between empathy and professional detachment, highlighting the risk of burnout and the need for effective coping strategies.

Training on empathic approaches and mutual support among colleagues were identified as key factors for navigating the emotional complexities of the job. Awareness and management of one’s emotions emerged as fundamental skills, with a direct impact on the quality of care and the well-being of nurses.

The identified thematic areas include the following:Situations of High Emotional Involvement: Descriptions of particularly impactful moments for nurses.Empathic Interventions: Actions undertaken by nurses in response to situations of strong emotional involvement.Perception and Sensations Related to Empathic Interventions: Reflections on the effects of empathic interventions, both personal and observed in others.Resistance to Emotional Involvement: Strategies adopted by nurses to manage or limit their emotional involvement.Emotional Awareness: The evolution of the ability to recognize and manage one’s emotions and those of others.

In Table 4, we inserted a summary of qualitative research findings with main themes, subthemes, and verbatim exemplifications.

Given the variety of work contexts of the respondents, the situations considered particularly engaging are varied. Among the main ones are accompanying death, young patients, patients who developed irreversible physical and psychological changes, abandonment of the elderly, patients from other cultures, and situations of discrimination or prejudice against patients. It is interesting to note that for some respondents, situations of high emotional involvement correspond to situations where their actions can help people, while others report high involvement in situations where they feel helpless.

**3.2.1. Theme 1**: The theme “Situations of High Emotional Involvement” highlights how nurses face significant emotional challenges in their diverse work contexts. Situations such as accompanying death, caring for young patients, and treating diseases with irreversible outcomes require deep emotional involvement, which can vary from a sense of usefulness to feelings of helplessness. In this theme, there are two subthemes:

(i) Assisting young patients: Caring for young patients in serious diagnostic procedures confronts nurses with vulnerability and emotional responsibility, often expressed as a personal challenge, “*this age group still gives me a bit of trouble… it affects me, you see*” (cod. 6.2).

(ii) Interaction with family members: Situations involving patients’ family members tend to intensify nurses’ emotional involvement, particularly when care extends beyond the patient to include their loved ones, “*it’s not that there isn’t involvement… but I actually don’t get as involved as when it involves family situations where the mom, children come in…*” (cod 4.18).

**3.2.2. Theme 2:** “*Empathic Interventions*”. The interventions implemented by nurses in highly emotionally involved situations vary greatly, reflecting the complexity and depth of their professional practice. In this theme, there are two subthemes:

(i) Identification with the patient: These interventions are often driven by a deep sense of empathy, which manifests in identifying with patients through similar personal experiences, such as empathizing with a patient’s parents or seeing a peer or one’s own child in a patient. This empathy is often at the core of nursing care, “*I would cry when a patient died because you see there a journey…*” (cod 10.14).

(ii) Support from colleagues and other figures: Colleague support proves essential, with nurses relying on coworkers for advice and emotional support. The importance of this support is highlighted by a nurse working in palliative care, “*I have always felt the support of the team*” (cod 7.20). This interaction among colleagues not only helps manage the emotional load but also strengthens team cohesion and the effectiveness of care.

**3.2.3. Theme 3:** The theme “*Perceptions and Sensations Related to Empathic Interventions*” explores how nurses perceive and emotionally react to empathic interventions, both those they carry out themselves and those performed by others, in contexts of high emotional involvement. Nurses’ reactions often focus on the emotions of family members and caregivers, as well as their personal reflections during and after the interventions. Some nurses expressed concerns about the risk of excessive involvement that can extend beyond the workplace and have personal repercussions. Reflections suggest that empathy is seen as a universal language that is necessary but must be adapted to the various cultural situations of patients to avoid misunderstandings or distances. Furthermore, criticism of colleagues perceived as lacking empathy shows the importance attributed to emotional sensitivity in the healthcare context. In this theme, there are two subthemes:

(i) Reflection on intense emotions: A nurse describes their emotional state in response to the suffering of a little girl and the difficulty of communicating with an unaware mother, “*… I didn’t know how to approach the mother…*” (cod. 15.7). This expresses the emotional complexity that nurses face in their role.

(ii) Considerations on empathy and professional training: Another nurse reflects on the importance of empathy and team support in managing emotionally charged situations, “*This brings you closer. It might be frightening, this risk of being scared, because you fear you cannot control it…*” (cod. 3.20). Here, the value of training and teamwork in facing emotional challenges is highlighted.

**3.2.4. Theme 4:** “*Resistance to emotional involvement*” in care situations is often seen as necessary to maintain professionalism in nursing. This resistance can develop following emotionally intense experiences that have not been fully processed, leaving nurses with feelings of incompleteness and frustration. Some nurses report that continuous exposure to emergency situations, where care practices must be rapid and technical, has led them to separate the emotional aspect from the technical to protect their emotional stability and maintain professional objectivity. In this theme, there are two subthemes:

(i) Excessive involvement to the detriment of professionalism: One nurse describes their approach to work, emphasizing the importance of maintaining a certain emotional distance, “*You are there to provide a service*” (cod. 6.12). This statement reflects the tension between being empathetic and the risk of losing professionalism if one allows emotions to take over.

(ii) Avoiding Excessive Involvement: In the context of pediatric anesthesia, a nurse explains how the speed of emergency procedures limits the emotional impact, “*you dedicate yourself only to the technical part*” (cod. 13.20). Eliminating stages that could cause emotional stress helps the nurse to focus exclusively on technical skills.

**3.2.5. Theme 5:** “*Emotional Awareness*” focuses on how nurses become aware of their own emotional states and those of others through experience and training. Interviews revealed that such awareness is crucial for effectively managing emotional involvement. More experienced nurses with many years of service exhibited a higher level of emotional awareness, showing how age and experience contribute to developing this skill. However, even less experienced nurses recognize the importance of learning how to manage their emotions, although they may not know exactly how to do so.

(i) Self-perceptions of the professional in relation to empathy: A nurse reflects on how experiences with certain types of patients and situations have honed their ability to manage their emotional responses, “*The experience with that type of patient, that type of situation has perhaps made me more capable of handling it…*” (cod. 7.18). This reflection shows how empathy evolves and intensifies with experience and age.

(ii) Evolution of empathic skills: Another nurse describes a long process of improving their empathic abilities over the years, “*there has always been a focus on refining the ability to connect and empathize with people*” (cod. 3.24). This statement highlights a continuous commitment to enhancing the empathic approach while maintaining emotional balance.

### 3.3. Data Triangulation

From the triangulation of results, it emerges that the drop in susceptibility levels in nurses under 29 years of age and with a service experience between 0 and 5 years that occurs with advancing age and service experience can be traced back to the macroareas identified in the qualitative analysis, specifically resistance to emotional involvement. In this macroarea, the theme identified and labeled as the loss of empathic abilities was identified in interview numbers 8-9-14-15, whose interviewees were, respectively, 40, 52, 31, and 30 years old. In the same way, the theme of avoiding emotional involvement was identified in interview numbers 4-8-9-11-12-13, whose interviewees were, respectively, 45, 40, 52, 24, 24, and 28 years old. In general, it can be stated that the data that emerged from the qualitative analysis are confirmed by the data that emerged from the qualitative research.

## 4. Discussion

This study aimed to evaluate empathy among nurses working in neuroscience contexts. Specifically, we measured the domains of susceptibility and resistance in nursing neuroscience contexts (quantitative study). Furthermore, we explored, in-depth, the empathetic experiences of nurses in neuroscience with their patients in clinical practice (qualitative study).

### 4.1. Discussion of the Results of the Quantitative Study

Quantitative findings highlight two relevant elements: (i) the risk of emotional exhaustion that occurs mainly in young people and in the first years of work; (ii) balancing susceptibility and resistance in nurses’ emotions in neuroscience.

#### 4.1.1. The Risk of Emotional Burnout in the Young and the Early Years of Service

The main findings of this study indicate that nurses in neuroscience exhibit a higher level of susceptibility, thus permeability and attention to the emotions of others, under the age of 29, then decreasing by at least half a point between the ages of 30 and 40, and remaining relatively stable in subsequent years. The same trend is observed concerning years of service, with a significant increase in susceptibility within the first 5 years of work. There are still many reservations in the literature regarding correlations between susceptibility, resistance, empathy, and sociodemographic data [53,54]. In agreement with Sommerlad, our study highlights that younger nurses tend to be more empathetic and emotionally involved compared with their older colleagues [55]. This could be attributed to greater emotional freshness, an initial enthusiasm for the chosen profession, and a lesser accumulation of traumatic experiences. Additionally, the early years of work report another level of susceptibility. These two data points support the potential risk of burnout, especially prevalent among young nurses and in the early years of the nursing profession, with consequent early abandonment of the profession, which is a highly relevant and studied aspect today due to the social and economic implications it determines [56]. Recent studies identify the risk of burnout as an already alarming phenomenon in nursing students and suggest interventions to facilitate the entry of young nurses into the clinic, as the early years of the profession are particularly at risk [57,58]. In the literature, nursing burnout is particularly linked to work contexts, such as critical care and palliative care [59,60,61]. However, even within neuroscience settings, nurses come into contact with critical care and palliative care situations, making it essential to seek solutions that can prevent or reduce this phenomenon.

#### 4.1.2. Balancing Susceptibility and Resistance in Nurses’ Emotions in Neuroscience

This study’s results indicate that both in years of service and age, susceptibility, after an initial increase, remains relatively stable over the years. Emotional resistance remains quite stable at around 5.5 points (on a Likert scale from 1 to 7) for all ages and years of service, with very minimal variations. Regarding years of service, the lines of susceptibility and resistance converge, with minimal differences, after 30 years of age and after the first 5 years of service. This result is interesting because it shows, with a high level of both susceptibility and resistance that persists over time, that the responding nurses perceive themselves as permeable to the emotions of patients and families and, at the same time, have developed a dimension of resistance, a defense against the stronger emotions that their work provokes, allowing them to continue to be empathetic [62]. However, these data may also suggest that with increasing experience, nurses develop defense mechanisms, such as avoidance and distancing, to protect themselves from emotional burnout. Considering only the data on high resistance might lead us to think that there is a reduction in empathy, which could compromise quality of care and therapeutic relationships with patients. However, the high level of susceptibility leads us to think that there is a balance between positive and negative emotions and that nurses, while maintaining a protective distance, are still able to be empathetic with patients and families, even in difficult situations.

### 4.2. Discussion of the Results of Qualitative Study

Qualitative results highlight the experiences of neuroscience nurses with empathy.

#### Empathy: The Experience of Neuroscience Nurses

Empathy is an essential construct for effective healthcare delivery [63]. When qualitative data are considered, neuroscience nurses claim to face difficult situations that have a strong emotional impact, but they also state that they are able to activate compensatory and adaptive strategies. Among these, we find the ability to reflect on the encountered situations [64,65] and also on their own emotions [18], the request for team support, the development of awareness on how to best manage their emotions in difficult situations, and the need to maintain a proper balance between involvement and detachment to be effective and competent professionals [18,64,65]. A factor that negatively affects the emotional compensation of nurses is the difficulty of relationships and understanding, often shown by managers and leaders, which emerged as a significant point in the interviews. Nurses report a sense of abandonment by administrations, which do not provide adequate support services to manage the stress associated with emotional involvement. In this aspect, the literature is rich and in agreement with the statements of our interviewees [66,67].

Nurses interviewed reported numerous situations where empathy played a crucial role in their professional activity. The most intense conditions reported involve accompanying patients to death, especially in cases of unfavorable prognosis or sudden death, as well as the emotional involvement towards young patients whose lives have been drastically altered by pathological events and patients with irreversible physical and mental changes. The qualitative data, in particular, showed how the respondents perceive empathy as a significant contribution to making care more humane and compassionate [63]. From the literature, the nurses participating in the study demonstrate that they recognize that a high level of empathy is fundamental for the quality of care provided [1,13]. Despite the positive aspects, the qualitative data reveal that a common strategy among nurses to manage emotional stress is to reduce emotional involvement. This strategy can manifest as a greater focus on the technical aspects of care, minimizing empathetic interaction with patients. This study’s results highlight how respondents can manage, with self-care strategies, reflection, and the development of awareness, the problems related to the intense emotional stresses they encounter at work [65,68]. In this direction, the study by Perez-Fuentes et al. identifies the work of healthcare personnel as characterized by high psychological and emotional demands and a high level of perceived stress. Therefore, promoting self-care and the well-being of nurses through strategies such as the development of awareness becomes essential to maintain optimal patient care [18].

### 4.3. General Discussion

#### 4.3.1. Data Integration Findings

The integrated qualitative–quantitative data showed an essential convergence on the themes of professionals’ ability to manage emotions and their ability to empathize with patients and families.

Neuroscientist nurses have shown that their empathy-related experiences are very similar to what the critical area nurses and palliative care nurses live. In fact, in the vastness and heterogeneity of neurological pathologies, we go from situations of assistance of medium intensity to conditions of criticism to accompanying situations at the end of life.

Qualitative data deepen and explain quantitative data and show that nurses can create deep connections with patients, demonstrating a genuine understanding of their sufferings and needs, but without showing obvious signs of burnout. The ability to resonate emotionally with patients facilitates not only the healing process but also the general well-being of the patient.

The ability, activated with experience, to be empathetic and protect themself promotes the maintenance of the professionals’ well-being [1,13].

#### 4.3.2. Implication for Policy and Practice

For the practices already implemented by respondents to protect themselves and remain empathetic, which we hypothesized are predominantly linked to a path of individual and professional growth and/or group experience, the literature suggests several other external support interventions, such as training [69,70], support in developing awareness with practices of meditation and mindfulness [71], and attention from superiors to the well-being of the professionals [41]. Perez-Fuentes et al. also specify that the introduction of development programs as a daily practice in the workplace improves the well-being of the professional and generates positive effects on the organization [18,63]. The improvement of the work climate and nursing action derive both from the progress of working conditions and the development of individual internal resources, which in turn facilitate the management of conditions of high stress, burnout, and emotional impact.

In view of the risk of burnout in young people during their first work experience, administrators are advised to set up safe pathways for recent graduates’ transition to work.

Administrators should also constantly measure the working well-being of professionals, especially those who operate in contexts with a high risk of compassion fatigue.

#### 4.3.3. Limitations

This study has some limitations.

The sample selected through a convenience approach may not be fully representative of all neuroscience settings or the nurses working in them. This limits the generalizability of the results. In addition, no statistically significant differences in the two analyzed variables (susceptibility and resistance) were found between the various care settings analyzed (neurosurgery units, neurology, etc.), which raises questions about the statistical power of this study given the sampling carried out. For future studies, the use of randomized sampling is envisaged so as to avoid the biases listed above.

## 5. Conclusions

In conclusion, our research confirms that empathic involvement (through a good balance of susceptibility and resistance) among nurses in neuroscience is an essential phenomenon for ensuring good-quality care and protecting the well-being of professionals. In this study, nurses recognize that they have developed strategies to regulate their emotions.

However, managers and administrations should constantly monitor this phenomenon and identify other strategies to support professionals.

The implications for clinical practice, education, and healthcare policy are significant, calling for greater investment in human and professional resources to strengthen empathy as a pillar of nursing care.

## Figures and Tables

**Figure 1 healthcare-12-02081-f001:**
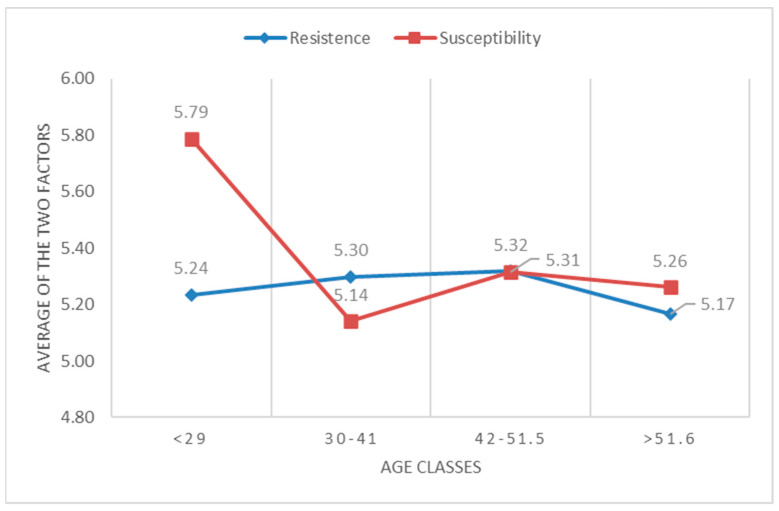
Trend of susceptibility and resistance based on registry age.

**Figure 2 healthcare-12-02081-f002:**
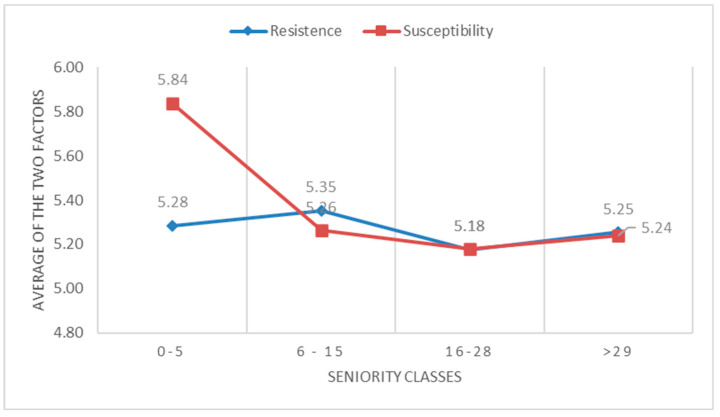
Trend in susceptibility and resistance based on years of service.

**Table 1 healthcare-12-02081-t001:** Quantitative Sample: Operational Unit (Numbers and Percentages).

Operational Unit	n	Percentage
Pediatrics	2	0.9%
Neuroradiology	4	1.9%
Neurorehabilitation	6	2.8%
Other	7	3.3%
Neurosurgery Operating Area	9	4.3%
1 (Unspecified Operational Unit)	10	4.7%
Nonclinical Area	16	7.6%
Mental Health	16	7.6%
Neurointensive Care	18	8.5%
Territory	21	10.0%
Neurosurgery	46	21.8%
Neurology	56	26.5%
Total	221	

**Table 2 healthcare-12-02081-t002:** Average resistance and susceptibility to emotional stimuli by care setting.

Care Settings	M Resistance	M Susceptibility
* NCC	5.3214	4.8267
Other	4.8673	5.6190
Neurosurgery	5.2795	5.6986
Operating Department NS **	4.6984	4.9852
Neurology	5.2870	5.4643
Nonclinical	5.3571	4.9167
Pediatrics	6.0357	5.7000
Mental health	5.1071	4.9917
Neuroradiology	4.9286	5.9500
Neurorehabilitation	4.9643	5.6111
Territory	5.4966	5.4857
Neurointensive Care	5.3016	5.0889

* NCC: Neurocritical Care. ** Operating Department NS: Neurosurgery.

**Table 3 healthcare-12-02081-t003:** Sociodemographic characteristics of participants.

Interview Code	Neuroscience Area	Age	Total Years of Work	Years in Neuroscience Field	Interview Duration
1	Neurointensive Care	24	2	1	24 min
2	Neurosurgery	59	39	25	42 min
3	Mental Health	54	35	32	55 min
4	Neurodegenerative	45	25	15	47 min
5	Mental Health	32	10	8	29 min
6	Neuroradiology	45	25	7	20 min
7	Neurodegenerative	45	24	10	33 min
8	Neurology	40	20	15	26 min
9	Neurology	52	33	25	23 min
10	Neurosurgery	46	22	8	33 min
11	Neurodegenerative	24	2	1	23 min
12	Neurosurgery	24	2	1	41 min
13	Neurosurgery OR *	28	7	2	39 min
14	Pediatrics	31	11	2	41 min
15	Pediatrics	30	12	10	26 min

* Neurosurgery; OR: Operation Room.

**Table 4 healthcare-12-02081-t004:** Summary of the results of the qualitative research.

Main Theme	Subtheme 1	Subtheme 2
**1. Situations of High Emotional Involvement**	**1.1.** **Assist young patients**	**1.2. Interactions with families**
	Verbatim *“The things that have touched me throughout my career are when I see… even in other diagnostics… MRI… rather than CT scans when these young kids come in… this age group still gives me a bit of trouble… well, it’s difficult… it affects me, you see”* (cod 6.2)	Verbatim *“I mean, it’s not that there isn’t involvement… but I actually don’t get as involved as when it involves family situations where the mom, children come in…”* (cod 4.18)
**2. Empathic Interventions**	**2.1. Identification with the patient**	**2.2. Support from colleagues and other figures**
	Verbatim *“I would cry when a patient died because you see there a journey… they are not just protocols (…) but they were patients, people we saw regularly, right? So when they were still strong, you fought with them and you fought until the end with them. I cried there, at the end of my work journey maybe I was tired…”* (cod 10.14)	Verbatim *“I have always felt a strong connection with the team, the supervision (…) the slightly larger group of the team that included the psychologist… if you needed it, but not particularly… it wasn’t so much the psychologist but rather the solidarity and listening from others, right?”* (cod 7.20)
**3. Perception and Sensations Related to Empathic Interventions**	**3.1. Internal reflection on the emotions felt**	**3.2. Reflections on behaviors**
	Verbatim *“A deep sorrow and a feeling of… I remember feeling frozen, you see, because I didn’t know how to approach the mother (…) and so also the difficulty of saying that the mother… you know… you don’t really know what the parent knew… and then this girl who was getting worse daily instead of better… it wasn’t easy”* (cod 15.7)	Verbatim *“So, what the person is experiencing, their here and now, their being in the world, and this brings you closer. It could be frightening, this risk of being scared, because you fear you cannot control it”* (cod 3.20)
**4. Resistance to Emotional Involvement**	**4.1.** **Excessive involvement to the detriment of professionalism**	**4.2. Avoiding excessive involvement**
	Verbatim *“It’s not really in nursing ethics to start crying behind the patient, otherwise you should change jobs. You are empathetic, you are understanding, but you don’t get involved to the extent of living their pain. You are there to provide a service”* (cod. 6.12)	Verbatim *“You carried forward what were then the technical maneuvers, so everything that was emotional was eliminated, is eliminated, and you dedicate yourself only to the technical part.”* (cod. 13.20)
**5. Emotional Awareness**	**5.1.** **Self-perceptions of the professional in relation to empathy**	**5.2.** **Evolution of empathic skills**
	Verbatim *“The experience with that type of patient, that type of situation, has perhaps made me more capable of handling it… but in reality, I still get moved, and I think that’s natural… then I must say that as I get older, because that is definitely happening, there’s more identification. So before, when I was caring for people, I could empathize with the patients”* (cod. 7.18)	Verbatim *“Over these 30 years, I’ve always worked on becoming more aware. There has always been a focus on refining my ability to connect and empathize with people, trying to help them in the most meaningful and impactful way possible, while managing to not get overwhelmed myself.”* (cod. 3.24)

## Data Availability

Data are available in an open repository in Mendeley: Mendeley Data, V1, doi: 10.17632/rmyf76tzym.1.

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
