# Peer review of "The Empathetic Involvement of Nurses in the Context of Neuroscience: A Mixed-Methods Study"

_healthcare, 2024, doi:10.3390/healthcare12202081_

Round 1

Reviewer 1 Report (Previous Reviewer 1)

Comments and Suggestions for Authors

Dear authors

I have reviewed the latest version of the manuscript with revisions and author responses indicated for each item I stated/suggested for the previous version. The final version of the manuscript has been sufficiently improved for Healthcare publication, and no further revisions are needed.

Author Response

Dear Reviewer,

Thank you very much!!

Best regards

Giovanna Artioli

Reviewer 2 Report (Previous Reviewer 2)

Comments and Suggestions for Authors

Dear authors

I would like to extend my congratulations to the authors for their dedication and commitment to the review process. Nevertheless, several queries remain unanswered. I will proceed to present my principal concerns.

1-     As this is a mixed methodology study, the authors must provide clarification in the introduction section regarding which variables will be analysed quantitatively and present a more robust rationale. Conversely, in the introduction session, it should be evident what the authors intend with the qualitative study, what the theoretical implications of this option are, and a robust rationale must be presented. On the other hand, the introduction section should be thought of as the “visiting card” of the study, making the purpose and contribution of the investigation clear. However, in its current state, the introduction does not serve its purpose given that it is not entirely clear what this work adds to the literature, why it is essential, what is already known in general terms, and how authors intend to contribute.

Additional comment: It is insufficient to justify the study on the grounds of its scarcity. It is incumbent upon authors to provide a clear and detailed account of the gap in the existing literature that they seek to address and to set forth their specific contributions to the field.

2-     It is not an optimal choice to situate the literature review within the introduction section, given the intricate nature of the study. It is therefore recommended that authors give due consideration to the separation of these two sections.

3-     In lines 142 to 158, the authors present a general explanation of the advantages of using a mixed methodology. These advantages are now widely recognized. It is anticipated that the authors will present a comprehensive and compelling rationale for the suitability of this methodology in the context of this study.

4-     The additional information revealed that the quantitative data was collected initially (for a period of 20 days), followed by the interviews. Furthermore, it became evident that the data were collected in close temporal proximity. Mixed methodology is an effective approach for addressing complex research questions that cannot be fully addressed by a single methodology. It is therefore incumbent on the authors to provide a clear and detailed justification for the use of a mixed methodology.

5-     In the discussion section, the authors should elucidate the main findings of the quantitative and qualitative studies.

Additional comment: It is recommended that authors create subheadings in the discussion section. 1. Discussion of the results of the quantitative study; 2. Discussion of qualitative data; 3. General discussion. Otherwise, the discussion is challenging to comprehend. The same procedure should be employed in the discussion of the implications of the study and its limitations.

Comments on the Quality of English Language

Some typing and punctuation errors have been found.

Author Response

Dear reviewer,

Thank you for your suggestions.

We inserted your comments and authors' responses in the file in the attachment.

Best regards

Giovanna Artioli

Reviewer 3 Report (Previous Reviewer 3)

Comments and Suggestions for Authors

Dear Authors,

Your manuscript explores the important area of ​​empathy in nursing within the specific context of neuroscience, which is a relatively under-researched area. The manuscript is relevant to the journal's readers because it explores empathy within the specific field of neuroscience in nursing, which is of great importance for professional development and clinical practice.

After the initial suggestions for correction in the article, it is obvious that you have implemented them, and I thank you for that.

Good luck with your further publication, best regards

Author Response

Dear reviewer,

Thank you very much!

Best regards

Giovanna Artioli

Round 2

Reviewer 2 Report (Previous Reviewer 2)

Comments and Suggestions for Authors

Dear authors,

I'm thrilled to congratulate you on your achievement!

I wish you all the best for continued success.

This manuscript is a resubmission of an earlier submission. The following is a list of the peer review reports and author responses from that submission.

Round 1

Reviewer 1 Report

Comments and Suggestions for Authors

Dear authors,

The current and significant topic of the manuscript addresses a challenging and under-researched issue. The study's unique contribution is in its provision of new data on empathic involvement, pointing to the essential link between professional development, nurses' well-being, and the quality of patient care. It also offers practical implications about the importance of investing in resources that would strengthen empathy among nurses in the clinical setting.

I would like to offer some suggestions to improve the manuscript:

-          Introduction: The introduction emphasizes the importance and concept of empathy and especially indicates the importance of maintaining empathy in clinical practice. Moreover, in lines 46-48, it is also indicated that research has shown that empathic interaction between a nurse and a patient leads to positive therapeutic effects. However, this quote needs to include more information, including the context and clinical environment in which this assessment was made. It would be helpful to supplement the data to get a more complete insight into the importance of empathy. Please provide these details.

-          Materials and Methods: Most of the data for a study that combined quantitative and qualitative to understand empathy within a nursing context comprehensively is correctly written. However, the statements in lines 110-111 must be supported by references. Since some socio-demographic data are mentioned later in the results, it would be significant to state in this part what all the socio-demographic data are and how they were collected. It would also be significant to provide more information about the criteria for inclusion in the study, whether all the nurses were from one institution or region (after the data in lines 134-136). I suggest you provide these details

-          Results: The results are presented in the quantitative and qualitative findings sections. My suggestions are as follows: For Table 1, it is necessary to match the table's name with the accompanying text in lines 171-172. The purpose of the Valid Percentage and Cumulative Percentage data remains to be determined; consider whether this data is necessary for you and in what context you have displayed it. For Table 2, it is necessary to match the table's name with the accompanying text in lines 180 -181. Also, please post explanations of the abbreviations in the footer below the table. In Figures 3 and 4, the axes must be clearly marked. Also, the names of these figures must be completed in accordance with the accompanying accompanying text. The title of Table 3 is also partially aligned with the accompanying text. It remains to be seen why you stated each participant's length of the interview. Did you expect differences? Were there any differences in relation to sociodemographic characteristics? Why is this data important to you? Think about it because you did not comment on them in the manuscript.

-          Discussion: The discussion is extensive, and the authors draw attention to many important questions raised by their research and the results of other authors. This is especially important because they discuss quantitative data in the context of common strategies for managing emotional stress. Discussing the qualitative data, they specifically focused on vulnerable groups of nurses and the impact of appropriate empathy training on the effectiveness of their services. At the end of the discussion section, it would be helpful to state the study's limitations.  Please provide these details.

-          Conclusion: Very general; be more specific, especially in relation to the set goals. Also, I suggest the conclusion be a separate section of the manuscript.

I hope you find my comments helpful.

Reviewer 2 Report

Comments and Suggestions for Authors

Dear authors

Thank you so much for your work. The manuscript presents an interesting topic. However, the study needs to be revised. I present my main concerns:

1-     As this is a mixed methodology study, the authors must provide clarification in the introduction section regarding which variables will be analysed quantitatively and present a more robust rationale. Conversely, in the introduction session, it should be evident what the authors intend with the qualitative study, what the theoretical implications of this option are, and a robust rationale must be presented. On the other hand, the introduction section should be thought of as the “visiting card” of the study, making the purpose and contribution of the investigation clear. However, in its current state, the introduction does not serve its purpose given that it is not entirely clear what this work adds to the literature, why it is essential, what is already known in general terms, and how authors intend to contribute.

2-     A literature review section is missing. This section is important because it is where authors must present the definitions of the variables under study and the arguments that support the relationships between variables.
On the other hand, the authors did not define the study hypotheses.
This is an unclear option for quantitative study.

3-     It is incumbent upon the authors to provide a more detailed account of the procedures employed in data collection. For instance, it would be advantageous to ascertain whether the data was gathered in person or online. It is important to ascertain whether the date mentioned in the text corresponds to the collection of quantitative or qualitative data. In addition, the approximate time required to complete the questionnaire and the time used to collect qualitative data should be indicated. Furthermore, the criteria used to select participants in the two studies (qualitative and quantitative) should be described. It is also essential for the authors to clarify the order used for these studies. In other words, it should be established whether the authors started with a qualitative or quantitative study.

4-     The authors in the abstract section mentioned the following: "This study helped explore the impact of empathy on nursing practices in neuroscience settings." In the method session, the authors mentioned the following: "The purpose of using the (BEES) in the research is to quantify the level of emotional empathy among nurses in the field of neuroscience, to examine how this influences their professional practice and their strategies for managing emotional stress". It is incumbent upon the authors to provide clarification regarding the specific practices and strategies they intend to analyze.

5-     With regard to the collection of qualitative data, it is essential that authors provide clarification regarding the methods employed to select the 15 participants. Was the interview conducted on an individual or group basis? What was the duration of the interview? What theoretical framework was used to inform the formulation of the questions presented?

6-     In the section entitled 'Setting and Sampling', it is evident that the participants were selected on the grounds of convenience. However, the authors do not provide clarification as to whether they are the participants in the quantitative study or the qualitative study.

7-     On page 4, the authors mentioned the following: "The quantitative collected data were statistically analyzed using SPSS® software. The goal was to identify trends and correlations within the sample through the Balanced Emotional Empathy Scale (BEES)”. When the definition of a study objective changes, it can be a bit confusing. Authors should be clear about what they are trying to achieve and keep to that throughout the study.

8-     It would be beneficial for authors to provide a more detailed description of the participants involved in the quantitative study.

9-     In the quantitative study, the authors did not perform confirmatory factor analysis, this issue must be justified, or presented as a limitation of the study.

10-  Figures 1 and 2 present information that was not substantiated in the introduction section.

11-  It was unclear whether theoretical saturation could be reached during the qualitative study.

12-  In the discussion section, the authors present the following objective: "This study aimed to explore nurses' responsiveness to others' emotional expressions and their ability to indirectly experience such emotional experiences in the field of neuroscience". Authors should clearly define the objective of the study.

13-  In the discussion section, the authors should elucidate the main findings of the quantitative and qualitative studies.

14-  The authors should add and substantiate the theoretical and practical contributions of the study. On the other hand, the authors did not present limitations and suggestions for future studies.

Reviewer 3 Report

Comments and Suggestions for Authors

Dear Authors,

Congratulations on the effort put into this research and the topic you have chosen. I have some tips for improving the quality of your manuscript, as follows:

In the Introduction Section streamline the introduction to focus on prior research specifically related to neuroscience and identify gaps that this study aims to fill. Avoid general discussions on empathy that do not directly lead to the study's objectives.

In the Materials and Methods section (Lines 133-139), provide a clearer explanation of the sampling method for the qualitative interviews and the quantitative survey. Include specific criteria used for selecting participants and detail any exclusion criteria to enhance the transparency and reproducibility of the study.

In the Qualitative Data Analysis (Lines 153-158) provide details on how data were coded, how themes were derived, the number of coders involved, and the methods used to ensure reliability and validity in the qualitative analysis.

The Discussion Section sometimes repeats information from the results without further analysis or synthesis. There is a lack of critical engagement with how findings interact with existing literature.

Good luck and kind regards
